# Sparse Embedded $k$-Means Clustering

**Weiwei Liu**[†,♮,∗] **Xiaobo Shen**[‡,∗]**, Ivor W. Tsang**[♮]

[†] School of Computer Science and Engineering, The University of New South Wales
[‡] School of Computer Science and Engineering, Nanyang Technological University
[♮] Centre for Artificial Intelligence, University of Technology Sydney
{liuweiwei863,njust.shenxiaobo}@gmail.com
ivor.tsang@uts.edu.au

## Abstract

The $k$-means clustering algorithm is a ubiquitous tool in data mining and machine learning that shows promising performance. However, its high computational cost has hindered its applications in broad domains. Researchers have successfully addressed these obstacles with dimensionality reduction methods. Recently, [1] develop a state-of-the-art random projection (RP) method for faster $k$-means clustering. Their method delivers many improvements over other dimensionality reduction methods. For example, compared to the advanced singular value decomposition based feature extraction approach, [1] reduce the running time by a factor of $\min\{n,d\}\epsilon^2 log(d)/k$ for data matrix $X \in \mathbb{R}^{n \times d}$ with $n$ data points and $d$ features, while losing only a factor of one in approximation accuracy. Unfortunately, they still require $\mathcal{O}(\frac{ndk}{\epsilon^2 log(d)})$ for matrix multiplication and this cost will be prohibitive for large values of $n$ and $d$. To break this bottleneck, we carefully build a sparse embedded $k$-means clustering algorithm which requires $\mathcal{O}(nnz(X))$ ($nnz(X)$ denotes the number of non-zeros in $X$) for fast matrix multiplication. Moreover, our proposed algorithm improves on [1]'s results for approximation accuracy by a factor of one. Our empirical studies corroborate our theoretical findings, and demonstrate that our approach is able to significantly accelerate $k$-means clustering, while achieving satisfactory clustering performance.

## 1 Introduction

Due to its simplicity and flexibility, the $k$-means clustering algorithm [2, 3, 4] has been identified as one of the top 10 data mining algorithms. It has shown promising results in various real world applications, such as bioinformatics, image processing, text mining and image analysis. Recently, the dimensionality and scale of data continues to grow in many applications, such as biological, finance, computer vision and web [5, 6, 7, 8, 9], which poses a serious challenge in designing efficient and accurate algorithmic solutions for $k$-means clustering.

To address these obstacles, one prevalent technique is dimensionality reduction, which embeds the original features into low dimensional space before performing $k$-means clustering. Dimensionality reduction encompasses two kinds of approaches: 1) feature selection (FS), which embeds the data into a low dimensional space by selecting the actual dimensions of the data; and 2) feature extraction (FE), which finds an embedding by constructing new artificial features that are, for example, linear combinations of the original features. Laplacian scores [10] and Fisher scores [11] are two basic feature selection methods. However, they lack a provable guarantee. [12] first propose a provable singular value decomposition (SVD) feature selection method. It uses the SVD to find $\mathcal{O}(k log(k/\epsilon)/\epsilon^2)$ actual features such that with constant probability the clustering structure

---

[∗]The first two authors make equal contributions.

Table 1: Dimension reduction methods for $k$-means clustering. The third column corresponds to the number of selected or extracted features. The fourth column corresponds to the time complexity of each dimension reduction method. The last column corresponds to the approximation accuracy. N/A denotes not available. $nnz(X)$ denotes the number of non-zeros in $X$. $\epsilon$ and $\delta$ represent the gap to optimality and the confidence level, respectively. Sparse embedding is abbreviated to SE.

| METHOD | DESCRIPTION | DIMENSIONS | TIME | ACCURACY |
|---|---|---|---|---|
| [13] | SVD-FE | $k$ | $\mathcal{O}(nd\min\{n,d\})$ | 2 |
| FOLKLORE | RP-FE | $\mathcal{O}(\frac{log(n)}{\epsilon^2})$ | $\mathcal{O}(\frac{ndlog(n)}{\epsilon^2 log(d)})$ | $1+\epsilon$ |
| [12] | SVD-FS | $\mathcal{O}(\frac{klog(k/\epsilon)}{\epsilon^2})$ | $\mathcal{O}(nd\min\{n,d\})$ | $2+\epsilon$ |
| [14] | SVD-FE | $\mathcal{O}(\frac{k}{\epsilon^2})$ | $\mathcal{O}(nd\min\{n,d\})$ | $1+\epsilon$ |
| [1] | RP-FE | $\mathcal{O}(\frac{k}{\epsilon^2})$ | $\mathcal{O}(\frac{ndk}{\epsilon^2 log(d)})$ | $2+\epsilon$ |
| [15] | RP-FE | $\mathcal{O}(\frac{log(n)}{n})$ | $\mathcal{O}(dlog(d)n + dlog(n))$ | N/A |
| THIS PAPER | SE-FE | $\mathcal{O}(\max(\frac{k+log(1/\delta)}{\epsilon^2}, \frac{6}{\epsilon^2\delta}))$ | $\mathcal{O}(nnz(X))$ | $1+\epsilon$ |

is preserved within a factor of $2+\epsilon$. [13] propose a popular feature extraction approach, where $k$ artificial features are constructed using the SVD such that the clustering structure is preserved within a factor of two. Recently, corollary 4.5 in [14]'s study improves [13]'s results, by claiming that $\mathcal{O}(\frac{k}{\epsilon^2})$ dimensions are sufficient for preserving $1+\epsilon$ accuracy.

Because SVD is computationally expensive, we focus on another important feature extraction method that randomly projects the data into low dimensional space. [1] develop a state-of-the-art random projection (RP) method, which is based on random sign matrices. Compared to SVD-based feature extraction approaches [14], [1] reduce the running time by a factor of $\min\{n,d\}\epsilon^2 log(d)/k^2$, while losing only a factor of one in approximation accuracy. They also improve the results of the folklore RP method by a factor of $log(n)/k$ in terms of the number of embedded dimensions and the running time, while losing a factor of one in approximation accuracy. Compared to SVD-based feature selection methods, [1] reduce the embedded dimension by a factor of $log(k/\epsilon)$ and the running time by a factor of $\min\{n,d\}\epsilon^2 log(d)/k$, respectively. Unfortunately, they still require $\mathcal{O}(\frac{ndk}{\epsilon^2 log(d)})$ for matrix multiplication and this cost will be prohibitive for large values of $n$ and $d$.

This paper carefully constructs a sparse matrix for the RP method that only requires $\mathcal{O}(nnz(X))$ for fast matrix multiplication. Our algorithm is significantly faster than other dimensionality reduction methods, especially when $nnz(X) << nd$. Theoretically, we show a provable guarantee for our algorithm. Given $\tilde{d} = \mathcal{O}(\max(\frac{k+log(1/\delta)}{\epsilon^2}, \frac{6}{\epsilon^2\delta}))$, with probability at least $1 - \mathcal{O}(\delta)$, our algorithm preserves the clustering structure within a factor of $1+\epsilon$, improving on the results of [12] and [1] by a factor of one for approximation accuracy. These results are summarized in Table 1.

Experiments on three real-world data sets show that our algorithm outperforms other dimension reduction methods. The results verify our theoretical analysis. We organize this paper as follows. Section 2 introduces the concept of $\epsilon$-approximation $k$-means clustering and our proposed sparse embedded $k$-means clustering algorithm. Section 3 analyzes the provable guarantee for our algorithm and experimental results are presented in Section 4. The last section provides our conclusions.

## 2 Sparse Embedded $k$-Means Clustering

### 2.1 $\epsilon$-Approximation $k$-Means Clustering

Given $X \in \mathbb{R}^{n \times d}$ with $n$ data points and $d$ features. We denote the transpose of the vector/matrix by superscript $'$ and the logarithms to base 2 by $log$. Let $r = rank(X)$. By using singular value decomposition (SVD), we have $X = U\Sigma V'$, where $\Sigma \in \mathbb{R}^{r \times r}$ is a positive diagonal matrix containing the singular values of $X$ in decreasing order ($\sigma_1 \geq \sigma_2 \geq \ldots \geq \sigma_r$), and $U \in \mathbb{R}^{n \times r}$ and $V \in \mathbb{R}^{d \times r}$ contain orthogonal left and right singular vectors of $X$. Let $U_k$ and $V_k$ represent $U$ and $V$ with all but their first $k$ columns zeroed out, respectively, and $\Sigma_k$ be $\Sigma$ with all but its largest $k$ singular values zeroed out. Assume $k \leq r$, [16] have already shown that $X_k = U_k\Sigma_k V_k'$ is the optimal rank $k$

approximation to $X$ for any unitarily invariant norm, including the Frobenius and spectral norms. The pseudoinverse of $X$ is given by $X^+ = V\Sigma^{-1}U'$. Assume $X_{r|k} = X - X_k$. $\mathbf{I}_n$ denotes the $n \times n$ identity matrix. Let $||X||_F$ be the Frobenius norm of matrix $X$. For concision, $||A||_2$ represents the spectral norm of $A$ if $A$ is a matrix and the Euclidean norm of $A$ if $A$ is a vector. Let $\text{nnz}(X)$ denote the number of non-zeros in $X$.

The task of $k$-means clustering is to partition $n$ data points in $d$ dimensions into $k$ clusters. Let $\mu_i$ be the centroid of the vectors in cluster $i$ and $c(x_i)$ be the cluster that $x_i$ is assigned to. Assume $D \in \mathbb{R}^{n \times k}$ is the indicator matrix which has exactly one non-zero element per row, which denotes cluster membership. The $i$-th data point belongs to the $j$-th cluster if and only if $D_{ij} = 1/\sqrt{z_j}$, where $z_j$ denotes the number of data points in cluster $j$. Note that $D'D = \mathbf{I}_k$ and the $i$-th row of $DD'X$ is the centroid of $x_i$'s assigned cluster. Thus, we have $\sum_{i=1}^{n} ||x_i - \mu_{c(x_i)}||_2^2 = ||X - DD'X||_F^2$. We formally define the $k$-means clustering task as follows, which is also studied in [12] and [1].

**Definition 1** ($k$-Means Clustering). *Given $X \in \mathbb{R}^{n \times d}$ and a positive integer $k$ denoting the number of clusters. Let $\mathcal{D}$ be the set of all $n \times k$ indicator matrices $D$. The task of $k$-means clustering is to solve*

$$\min_{D \in \mathcal{D}} ||X - DD'X||_F^2 \tag{1}$$

To accelerate the optimization of problem 1, we aim to find a $\epsilon$-approximate solution for problem 1 by optimizing $D$ (either exactly or approximately) over an embedded matrix $\hat{X} \in \mathbb{R}^{n \times \tilde{d}}$ with $\tilde{d} < d$. To measure the quality of approximation, we first define the $\epsilon$-approximation embedded matrix:

**Definition 2** ($\epsilon$-Approximation Embedded Matrix). *Given $0 \le \epsilon < 1$ and a non-negative constant $\tau$. $\hat{X} \in \mathbb{R}^{n \times \tilde{d}}$ with $\tilde{d} < d$ is a $\epsilon$-approximation embedded matrix for $X$, if*

$$(1 - \epsilon)||X - DD'X||_F^2 \le ||\hat{X} - DD'\hat{X}||_F^2 + \tau \le (1 + \epsilon)||X - DD'X||_F^2 \tag{2}$$

We show that a $\epsilon$-approximation embedded matrix is sufficient for approximately optimizing problem 1:

**Lemma 1** ($\epsilon$-Approximation $k$-Means Clustering). *Given $X \in \mathbb{R}^{n \times d}$ and $\mathcal{D}$ be the set of all $n \times k$ indicator matrices $D$, let $D^* = \arg\min_{D \in \mathcal{D}} ||X - DD'X||_F^2$. Given $\hat{X} \in \mathbb{R}^{n \times \tilde{d}}$ with $\tilde{d} < d$, let $\hat{D}^* = \arg\min_{D \in \mathcal{D}} ||\hat{X} - DD'\hat{X}||_F^2$. If $\hat{X}$ is a $\epsilon'$-approximation embedded matrix for $X$, given $\epsilon = 2\epsilon'/(1 - \epsilon')$, then for any $\gamma \ge 1$, if $||\hat{X} - \hat{D}\hat{D}'\hat{X}||_F^2 \le \gamma ||\hat{X} - \hat{D}^*\hat{D}^{*'}\hat{X}||_F^2$, we have*

$$||X - \hat{D}\hat{D}'X||_F^2 \le (1 + \epsilon)\gamma ||X - D^*D^{*'}X||_F^2$$

*Proof.* By definition, we have $||\hat{X} - \hat{D}^*\hat{D}^{*'}\hat{X}||_F^2 \le ||\hat{X} - D^*D^{*'}\hat{X}||_F^2$ and thus

$$||\hat{X} - \hat{D}\hat{D}'\hat{X}||_F^2 \le \gamma ||\hat{X} - D^*D^{*'}\hat{X}||_F^2 \tag{3}$$

Since $\hat{X}$ is a $\epsilon$-approximation embedded matrix for $X$, we have

$$||\hat{X} - D^*D^{*'}\hat{X}||_F^2 \le (1 + \epsilon')||X - D^*D^{*'}X||_F^2 - \tau$$
$$||\hat{X} - \hat{D}\hat{D}'\hat{X}||_F^2 \ge (1 - \epsilon')||X - \hat{D}\hat{D}'X||_F^2 - \tau \tag{4}$$

Combining Eq.(3) and Eq.(4), we obtain:

$$(1 - \epsilon')||X - \hat{D}\hat{D}'X||_F^2 - \tau \le ||\hat{X} - \hat{D}\hat{D}'\hat{X}||_F^2 \le \gamma ||\hat{X} - D^*D^{*'}\hat{X}||_F^2$$
$$\le (1 + \epsilon')\gamma ||X - D^*D^{*'}X||_F^2 - \tau\gamma \tag{5}$$

Eq.(5) implies that

$$||X - \hat{D}\hat{D}'X||_F^2 \le (1 + \epsilon')/(1 - \epsilon')\gamma ||X - D^*D^{*'}X||_F^2 \le (1 + \epsilon)\gamma ||X - D^*D^{*'}X||_F^2 \tag{6}$$

$\square$

**Remark.** Lemma 1 implies that if $\hat{D}$ is an optimal solution for $\hat{X}$, then it also preserves $\epsilon$-approximation for $X$. Parameter $\gamma$ allows $\hat{D}$ to be approximately global optimal for $\hat{X}$.

---
**Algorithm 1** Sparse Embedded $k$-Means Clustering
---
**Input:** $X \in \mathbb{R}^{n \times d}$. Number of clusters $k$.
**Output:** $\epsilon$-approximate solution for problem 1.
 1: Set $\tilde{d} = \mathcal{O}(\max(\frac{k+log(1/\delta)}{\epsilon^2}, \frac{6}{\epsilon^2 \delta}))$.
 2: Build a random map $h$ so that for any $i \in [d]$, $h(i) = j$ for $j \in [\tilde{d}]$ with probability $1/\tilde{d}$.
 3: Construct matrix $\Phi \in \{0, 1\}^{d \times \tilde{d}}$ with $\Phi_{i, h(i)} = 1$, and all remaining entries 0.
 4: Construct matrix $Q \in \mathbb{R}^{d \times d}$ is a random diagonal matrix whose entries are i.i.d. Rademacher variables.
 5: Compute the product $\hat{X} = XQ\Phi$ and run exact or approximate $k$-means algorithms on $\hat{X}$.
---

## 2.2 Sparse Embedding

[1] construct a random embedded matrix for fast $k$-means clustering by post-multiplying $X$ with a $d \times \tilde{d}$ random matrix having entries $\frac{1}{\sqrt{\tilde{d}}}$ or $\frac{-1}{\sqrt{\tilde{d}}}$ with equal probability. However, this method requires $\mathcal{O}(\frac{ndk}{\epsilon^2 log(d)})$ for matrix multiplication and this cost will be prohibitive for large values of $n$ and $d$. To break this bottleneck, Algorithm 1 demonstrates our sparse embedded $k$-means clustering, which requires $\mathcal{O}(nnz(X))$ for fast matrix multiplication. Our algorithm is significantly faster than other dimensionality reduction methods, especially when $nnz(X) << nd$. For an index $i$ taking values in the set $\{1, \cdots, n\}$, we write $i \in [n]$.

Next, we state our main theorem to show that $XQ\Phi$ is the $\epsilon$-approximation embedded matrix for $X$:

**Theorem 1.** *Let $\Phi$ and $Q$ be constructed as in Algorithm 1 and $R = (Q\Phi)' \in \mathbb{R}^{\tilde{d} \times d}$. Given $\tilde{d} = \mathcal{O}(\max(\frac{k+log(1/\delta)}{\epsilon^2}, \frac{6}{\epsilon^2 \delta}))$. Then for any $X \in \mathbb{R}^{n \times d}$, with a probability of at least $1 - \mathcal{O}(\delta)$, $XR'$ is the $\epsilon$-approximation embedded matrix for $X$.*

# 3 Proofs

Let $Z = \mathbf{I}_n - DD'$ and $tr$ be the trace notation. Eq.(2) can be formulated as: $(1 - \epsilon)tr(ZXX'Z) \leq tr(Z\hat{X}\hat{X}'Z) + \tau \leq (1+\epsilon)tr(ZXX'Z)$. Then, we try to approximate $XX'$ with $\hat{X}\hat{X}'$. To prove our main theorem, we write $\hat{X} = XR'$ and our goal is to show that $tr(ZXX'Z)$ can be approximated by $tr(ZXR'RX'Z)$. Lemma 2 provides conditions on the error matrix $\mathscr{E} = \hat{X}\hat{X}' - XX'$ that are sufficient to guarantee that $\hat{X}$ is a $\epsilon$-approximation embedded matrix for $X$. For any two symmetric matrices $A, B \in \mathbb{R}^{n \times n}$, $A \preceq B$ indicates that $B - A$ is positive semidefinite. Let $\lambda_i(A)$ denote the $i$-th largest eigenvalue of $A$ in absolute value. $\langle \cdot, \cdot \rangle$ represents the inner product, and $\mathbf{0}_{n \times d}$ denotes an $n \times d$ zero matrix with all its entries being zero.

**Lemma 2.** *Let $C = XX'$ and $\hat{C} = \hat{X}\hat{X}'$. If we write $\hat{C} = C + \mathscr{E}_1 + \mathscr{E}_2 + \mathscr{E}_3 + \mathscr{E}_4$, where:*

   *(i) $\mathscr{E}_1$ is symmetric and $-\epsilon_1 C \preceq \mathscr{E}_1 \preceq \epsilon_1 C$.*

   *(ii) $\mathscr{E}_2$ is symmetric, $\sum_{i=1}^{k} |\lambda_i(\mathscr{E}_2)| \leq \epsilon_2 ||X_{r|k}||_F^2$, and $tr(\mathscr{E}_2) \leq \tilde{\epsilon}_2 ||X_{r|k}||_F^2$.*

   *(iii) The columns of $\mathscr{E}_3$ fall in the column span of $C$ and $tr(\mathscr{E}_3' C^+ \mathscr{E}_3) \leq \epsilon_3^2 ||X_{r|k}||_F^2$.*

   *(iv) The rows of $\mathscr{E}_4$ fall in the row span of $C$ and $tr(\mathscr{E}_4 C^+ \mathscr{E}_4') \leq \epsilon_4^2 ||X_{r|k}||_F^2$.*

*and $\epsilon_1 + \epsilon_2 + \tilde{\epsilon}_2 + \epsilon_3 + \epsilon_4 = \epsilon$, then $\hat{X}$ is a $\epsilon$-approximation embedded matrix for $X$. Specifically, we have $(1 - \epsilon)tr(ZCZ) \leq tr(Z\hat{C}Z) - \min\{0, tr(\mathscr{E}_2)\} \leq (1 + \epsilon)tr(ZCZ)$.*

The proof can be referred to [17]. Next, we show $XR'$ is the $\epsilon$-approximation embedded matrix for $X$. We first present the following theorem:

**Theorem 2.** *Assume $r > 2k$ and let $V_{2k} \in \mathbb{R}^{d \times r}$ represent $V$ with all but their first $2k$ columns zeroed out. We define $M_1 = V_{2k}'$, $M_2 = \sqrt{k}/||X_{r|k}||_F (X - XV_{2k}V_{2k}')$ and $M \in \mathbb{R}^{(n+r) \times d}$ as containing $M_1$ as its first $r$ rows and $M_2$ as its lower $n$ rows. We construct $R = (Q\Phi)' \in \mathbb{R}^{\tilde{d} \times d}$,*

which is shown in Algorithm 1. Given $\tilde{d} = \mathcal{O}(\max(\frac{k+log(1/\delta)}{\epsilon^2}, \frac{6}{\epsilon^2\delta}))$, then for any $X \in \mathbb{R}^{n \times d}$, with a probability of at least $1 - \mathcal{O}(\delta)$, we have

(i) $||(RM')'(RM') - MM'||_2 < \epsilon$.

(ii) $|\,||RM_2'||_F^2 - ||M_2'||_F^2| \le \epsilon k$.

*Proof.* To prove the first result, one can easily check that $M_1 M_2' = \mathbf{0}_{r \times n}$, thus $MM'$ is a block diagonal matrix with an upper left block equal to $M_1 M_1'$ and lower right block equal to $M_2 M_2'$. The spectral norm of $M_1 M_1'$ is 1. $||M_2 M_2'||_2 = ||M_2||_2^2 = \frac{k||X - XV_{2k}V_{2k}'||_2^2}{||X_{r|k}||_F^2} = \frac{k||X_{r|2k}||_2^2}{||X_{r|k}||_F^2}$. As $||X_{r|k}||_F^2 \ge k||X_{r|2k}||_2^2$, we derive $||M_2 M_2'||_2 \le 1$. Since $MM'$ is a block diagonal matrix, we have $||M||_2^2 = ||MM'||_2 = \max\{||M_1 M_1'||_2, ||M_2 M_2'||_2\} = 1$. $tr(M_1 M_1') = 2k$. $tr(M_2 M_2') = \frac{k||X_{r|2k}||_F^2}{||X_{r|k}||_F^2}$. As $||X_{r|k}||_F^2 \ge ||X_{r|2k}||_F^2$, we derive $tr(M_2 M_2') \le k$. Then we have $||M||_F^2 = tr(MM') = tr(M_1 M_1') + tr(M_2 M_2') \le 3k$. Applying Theorem 6 from [18], we can obtain that given $\tilde{d} = \mathcal{O}(\frac{k+log(1/\delta)}{\epsilon^2})$, with a probability of at least $1 - \delta$, $||(RM')'(RM') - MM'||_2 < \epsilon$.

The proof of the second result can be found in the Supplementary Materials. $\square$

Based on Theorem 2, we show that $\hat{X} = XR'$ satisfies the conditions of Lemma 2.

**Lemma 3.** *Assume $r > 2k$ and we construct $M$ and $R$ as in Theorem 2. Given $\tilde{d} = \mathcal{O}(\max(\frac{k+log(1/\delta)}{\epsilon^2}, \frac{6}{\epsilon^2\delta}))$, then for any $X \in \mathbb{R}^{n \times d}$, with a probability of at least $1 - \mathcal{O}(\delta)$, $\hat{X} = XR'$ satisfies the conditions of Lemma 2.*

*Proof.* We construct $H_1 \in \mathbb{R}^{n \times (n+r)}$ as $H_1 = [XV_{2k}, \mathbf{0}_{n \times n}]$, thus $H_1 M = XV_{2k}V_{2k}'$. And we set $H_2 \in \mathbb{R}^{n \times (n+r)}$ as $H_2 = [\mathbf{0}_{n \times r}, \frac{||X_{r|k}||_F}{\sqrt{k}} \mathbf{I}_n]$, so we have $H_2 M = \frac{||X_{r|k}||_F}{\sqrt{k}} M_2 = X - XV_{2k}V_{2k}' = X_{r|2k}$ and $X = H_1 M + H_2 M$ and we obtain the following:

$$\mathcal{E} = \hat{X}\hat{X}' - XX' = XR'RX' - XX' = ① + ② + ③ + ④ \tag{7}$$

Where $① = H_1 M R' R M' H_1' - H_1 M M' H_1'$, $② = H_2 M R' R M' H_2' - H_2 M M' H_2'$, $③ = H_1 M R' R M' H_2' - H_1 M M' H_2'$ and $④ = H_2 M R' R M' H_1' - H_2 M M' H_1'$. We bound ①, ②, ③ and ④ separately, showing that each corresponds to one of the error terms described in Lemma 2.

**Bounding ①.**

$$\mathcal{E}_1 = H_1 M R' R M' H_1' - H_1 M M' H_1' = XV_{2k}V_{2k}'R'RV_{2k}V_{2k}'X' - XV_{2k}V_{2k}'V_{2k}V_{2k}'X' \tag{8}$$

$\mathcal{E}_1$ is symmetric. By Theorem 2, we know that with a probability of at least $1 - \delta$, $||(RM')'(RM') - MM'||_2 < \epsilon$ holds. Then we get $-\epsilon \mathbf{I}_{n+r} \preceq (RM')'(RM') - MM' \preceq \epsilon \mathbf{I}_{n+r}$. And we derive the following:

$$-\epsilon H_1 H_1' \preceq \mathcal{E}_1 \preceq \epsilon H_1 H_1' \tag{9}$$

For any vector $v$, $v'XV_{2k}V_{2k}'V_{2k}V_{2k}'X'v = ||V_{2k}V_{2k}'X'v||_2^2 \le ||V_{2k}V_{2k}'||_2^2||X'v||_2^2 = ||X'v||_2^2 = v'XX'v$, so $H_1 M M' H_1' = XV_{2k}V_{2k}'V_{2k}V_{2k}'X' \preceq XX'$. Since $H_1 M M' H_1' = XV_{2k}V_{2k}'V_{2k}V_{2k}'X' = XV_{2k}V_{2k}'X' = H_1 H_1'$, we have

$$H_1 H_1' = H_1 M M' H_1' \preceq XX' = C \tag{10}$$

Combining Eqs.(9) and (10), we arrive at a probability of at least $1 - \delta$,

$$-\epsilon C \preceq \mathcal{E}_1 \preceq \epsilon C \tag{11}$$

satisfying the first condition of Lemma 2.

**Bounding ②.**

$$\begin{aligned}
\mathcal{E}_2 &= H_2 M R' R M' H_2' - H_2 M M' H_2' \\
&= (X - XV_{2k}V_{2k}')R'R(X - XV_{2k}V_{2k}')' - (X - XV_{2k}V_{2k}')(X - XV_{2k}V_{2k}')'
\end{aligned} \tag{12}$$

$\mathscr{E}_2$ is symmetric. Note that $H_2$ just selects $M_2$ from $M$ and scales it by $||X_{r|k}||_F/\sqrt{k}$. Using Theorem 2, we know that with a probability of at least $1 - \delta$,

$$tr(\mathscr{E}_2) = \frac{||X_{r|k}||_F^2}{k}tr(M_2R'RM_2' - M_2M_2') \le \epsilon||X_{r|k}||_F^2 \tag{13}$$

Applying Theorem 6.2 from [19] and rescaling $\epsilon$, we can obtain a probability of at least $1 - \delta$,

$$||\mathscr{E}_2||_F = ||X_{r|2k}R'RX_{r|2k}' - X_{r|2k}X_{r|2k}'||_F \le \frac{\epsilon}{\sqrt{k}}||X_{r|2k}||_F^2 \tag{14}$$

Combining Eq.(14), Cauchy-Schwarz inequality and $||X_{r|2k}||_F^2 \le ||X_{r|k}||_F^2$, we get that with a probability of at least $1 - \delta$,

$$\sum_{i=1}^{k} |\lambda_i(\mathscr{E}_2)| \le \sqrt{k}||\mathscr{E}_2||_F \le \epsilon||X_{r|k}||_F^2 \tag{15}$$

Eqs.(13) and (15) satisfy the second conditions of Lemma 2.

**Bounding ③.**

$$\begin{aligned}\mathscr{E}_3 &= H_1MR'RM'H_2' - H_1MM'H_2' \\ &= XV_{2k}V_{2k}'R'R(X - XV_{2k}V_{2k}')' - XV_{2k}V_{2k}'(X - XV_{2k}V_{2k}')'\end{aligned} \tag{16}$$

The columns of $\mathscr{E}_3$ are in the column span of $H_1M = XV_{2k}V_{2k}'$, and so in the column span of $C$. $||V_{2k}||_F^2 = tr(V_{2k}'V_{2k}) = 2k$. As $V_{2k}'V = V_{2k}'V_{2k}$, $V_{2k}'X_{r|2k}' = V_{2k}'(V\Sigma U' - V_{2k}\Sigma_{2k}U_{2k}') = \Sigma_{2k}U_{2k}' - \Sigma_{2k}U_{2k}' = \mathbf{0}_{r \times n}$. Applying Theorem 6.2 from [19] again and rescaling $\epsilon$, we can obtain that with a probability of at least $1 - \delta$,

$$\begin{aligned}tr(\mathscr{E}_3'C^+\mathscr{E}_3) &= ||\Sigma^{-1}U'(H_1MR'RM'H_2' - H_1MM'H_2')||_F^2 \\ &= ||V_{2k}'R'RX_{r|2k}' - \mathbf{0}_{r \times n}||_F^2 \le \epsilon^2||X_{r|k}||_F^2\end{aligned} \tag{17}$$

Thus, Eq.(17) satisfies the third condition of Lemma 2.

**Bounding ④.**

$$\begin{aligned}\mathscr{E}_4 &= H_2MR'RM'H_1' - H_2MM'H_1' \\ &= (X - XV_{2k}V_{2k}')R'RV_{2k}V_{2k}'X' - (X - XV_{2k}V_{2k}')V_{2k}V_{2k}'X'\end{aligned} \tag{18}$$

$\mathscr{E}_4 = \mathscr{E}_3'$ and thus we immediately have that with a probability of at least $1 - \delta$,

$$tr(\mathscr{E}_4C^+\mathscr{E}_4') \le \epsilon^2||X_{r|k}||_F^2 \tag{19}$$

Lastly, Eqs.(11), (13), (15), (17) and (19) ensure that, for any $X \in \mathbb{R}^{n \times d}$, $\hat{X} = XR'$ satisfies the conditions of Lemma 2 and is the $\epsilon$-approximation embedded matrix for $X$ with a probability of at least $1 - \mathcal{O}(\delta)$.  □

# 4 Experiment

## 4.1 Data Sets and Baselines

We denote our proposed sparse embedded $k$-means clustering algorithm as SE for short. This section evaluates the performance of the proposed method on four real-world data sets: COIL20, SECTOR, RCV1 and ILSVRC2012. The COIL20 [20] and ILSVRC2012 [21] data sets are collected from website[34], and other data sets are collected from the LIBSVM website[5]. The statistics of these data sets are presented in the Supplementary Materials.

We compare SE with several other dimensionality reduction techniques:

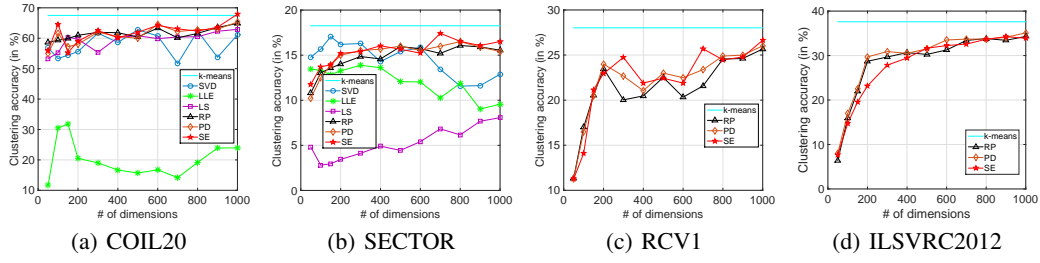

Figure 1: Clustering accuracy of various methods on COIL20, SECTOR, RCV1 and ILSVRC2012 data sets.

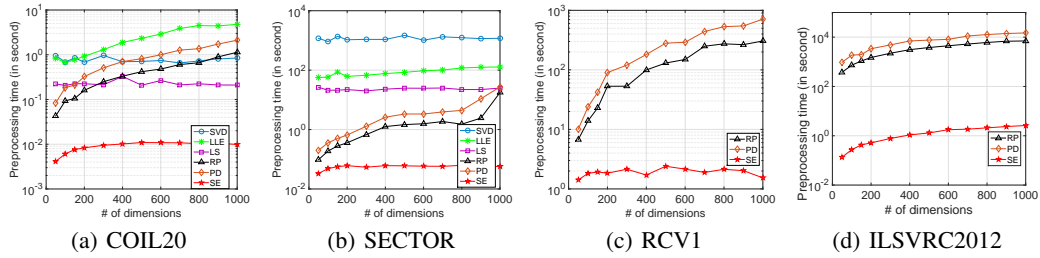

Figure 2: Dimension reduction time of various methods on COIL20, SECTOR, RCV1 and ILSVRC2012 data sets.

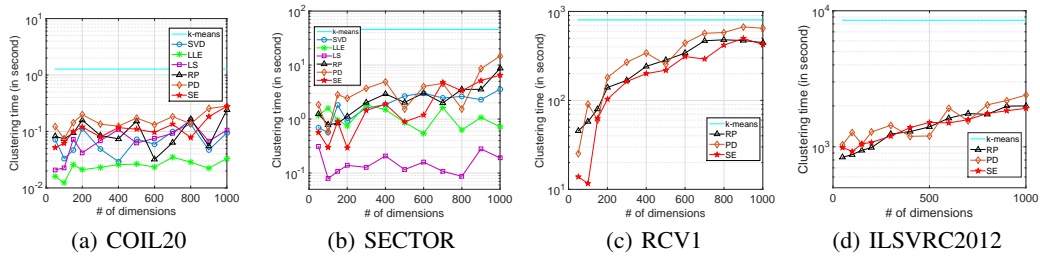

Figure 3: Clustering time of various methods on COIL20, SECTOR, RCV1 and ILSVRC2012 data sets.

- SVD: The singular value decomposition or principal components analysis dimensionality reduction approach.
- LLE: The local linear embedding (LLE) algorithm is proposed by [22]. We use the code from website[6] with default parameters.
- LS: [10] develop the laplacian score (LS) feature selection method. We use the code from website[7] with default parameters.
- PD: [15] propose an advanced compression scheme for accelerating $k$-means clustering. We use the code from website[8] with default parameters.
- RP: The state-of-the-art random projection method is proposed by [1].

After dimensionality reduction, we run all methods on a standard $k$-means clustering package, which is from website[9] with default parameters. We also compare all these methods against the standard $k$-means algorithm on the full dimensional data sets. To measure the quality of all methods, we report clustering accuracy based on the labelled information of the input data. Finally, we report the running

times (in seconds) of both the dimensionality reduction procedure and the $k$-means clustering for all baselines.

## 4.2 Results

The experimental results of various methods on all data sets are shown in Figures 1, 2 and 3. The $Y$ axes of Figures 2 and 3 represent dimension reduction and clustering time in log scale. We can't get the results of SVD, LLE and LS within three days on RCV1 and ILSVRC2012 data sets. Thus, these results are not reported.

From Figures 1, 2 and 3, we can see that:

- As the number of embedded dimensions increases, the clustering accuracy and running times of all dimensionality reduction methods increases, which is consistent with the empirical results in [1].

- Our proposed dimensionality reduction method has superior performance compared to the RP method and other baselines in terms of accuracy, which verifies our theoretical results. LLE and LS generally underperforms on the COIL20 and SECTOR data sets.

- SVD and LLE are the two slowest methods compared with the other baselines in terms of dimensionality reduction time. The dimension reduction time of the RP method increases significantly with the increasing dimensions, while our method obtains a stable and lowest dimensionality reduction time. We achieve several hundred orders of magnitude faster than the RP method and other baselines. The results also support our complexity analysis.

- All dimensionality reduction methods are significantly faster than standard $k$-means algorithm with full dimensions. Finally, we conclude that our proposed method is able to significantly accelerate $k$-means clustering, while achieving satisfactory clustering performance.

## 5   Conclusion

The $k$-means clustering algorithm is a ubiquitous tool in data mining and machine learning with numerous applications. The increasing dimensionality and scale of data has provided a considerable challenge in designing efficient and accurate $k$-means clustering algorithms. Researchers have successfully addressed these obstacles with dimensionality reduction methods. These methods embed the original features into low dimensional space, and then perform $k$-means clustering on the embedded dimensions. SVD is one of the most popular dimensionality reduction methods. However, it is computationally expensive. Recently, [1] develop a state-of-the-art RP method for faster $k$-means clustering. Their method delivers many improvements over other dimensionality reduction methods. For example, compared to an advanced SVD-based feature extraction approach [14], [1] reduce the running time by a factor of $\min\{n, d\}\epsilon^2 log(d)/k$, while only losing a factor of one in approximation accuracy. They also improve the result of the folklore RP method by a factor of $log(n)/k$ in terms of the number of embedded dimensions and the running time, while losing a factor of one in approximation accuracy. Unfortunately, it still requires $\mathcal{O}(\frac{ndk}{\epsilon^2 log(d)})$ for matrix multiplication and this cost will be prohibitive for large values of $n$ and $d$. To break this bottleneck, we carefully construct a sparse matrix for the RP method that only requires $\mathcal{O}(nnz(X))$ for fast matrix multiplication. Our algorithm is significantly faster than other dimensionality reduction methods, especially when $nnz(X) << nd$. Furthermore, we improve the results of [12] and [1] by a factor of one for approximation accuracy. Our empirical studies demonstrate that our proposed algorithm outperforms other dimension reduction methods, which corroborates our theoretical findings.

**Acknowledgments**

We would like to thank the area chairs and reviewers for their valuable comments and constructive suggestions on our paper. This project is supported by the ARC Future Fellowship FT130100746, ARC grant LP150100671, DP170101628, DP150102728, DP150103071, NSFC 61232006 and NSFC 61672235.

## Footnotes

[2] Refer to Section 2.1 for the notations.

[3]http://www.cs.columbia.edu/CAVE/software/softlib/coil-20.php

[4]http://www.image-net.org/challenges/LSVRC/2012/

[5]https://www.csie.ntu.edu.tw/ cjlin/libsvmtools/datasets/

[6]http://www.cs.nyu.edu/ roweis/lle/

[7]www.cad.zju.edu.cn/home/dengcai/Data/data.html

[8]https://github.com/stephenbeckr/SparsifiedKMeans

[9]www.cad.zju.edu.cn/home/dengcai/Data/data.html

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
