[Supplementary Material · SE_Supplementary.pdf]

# Sparse Embedded $k$-Means Clustering
## (Supplementary)

**Weiwei Liu**[†,♮,∗] **Xiaobo Shen**[‡,∗], **Ivor W. Tsang**[♮]
[†] School of Computer Science and Engineering, The University of New South Wales
[‡] School of Computer Science and Engineering, Nanyang Technological University
[♮] Centre for Artificial Intelligence, University of Technology Sydney
{liuweiwei863,njust.shenxiaobo}@gmail.com
ivor.tsang@uts.edu.au

## Abstract

In this supplementary file, we first present the proof of Theorem 2 in the main paper. After that, we present the statistics on the data sets used in the main paper, and a case study.

## 1 Proof of Theorem 2

Let $Z = \mathbf{I}_n - DD'$ and $tr$ be the trace notation. Eq.(2) in the main paper can be formulated as: $(1 - \epsilon)tr(ZXX'Z) \leq tr(Z\hat{X}\hat{X}'Z) + \tau \leq (1 + \epsilon)tr(ZXX'Z)$. Then, we try to approximate $XX'$ with $\hat{X}\hat{X}'$. To prove our main theorem, we write $\hat{X} = XR'$ and our goal is to show that $tr(ZXX'Z)$ can be approximated by $tr(ZXR'RX'Z)$. Lemma 2 provides conditions on the error matrix $\mathscr{E} = \hat{X}\hat{X}' - XX'$ that are sufficient to guarantee that $\hat{X}$ is a $\epsilon$-approximation embedded matrix for $X$. For any two symmetric matrices $A, B \in \mathbb{R}^{n \times n}$, $A \preceq B$ indicates that $B - A$ is positive semidefinite. Let $\lambda_i(A)$ denote the $i$-th largest eigenvalue of $A$ in absolute value. $\langle \cdot, \cdot \rangle$ represents the inner product, and $\mathbf{0}_{n \times d}$ denotes an $n \times d$ zero matrix with all its entries being zero.

**Lemma 2.** *Let $C = XX'$ and $\hat{C} = \hat{X}\hat{X}'$. If we write $\hat{C} = C + \mathscr{E}_1 + \mathscr{E}_2 + \mathscr{E}_3 + \mathscr{E}_4$, where:*

  *(i) $\mathscr{E}_1$ is symmetric and $-\epsilon_1 C \preceq \mathscr{E}_1 \preceq \epsilon_1 C$.*

  *(ii) $\mathscr{E}_2$ is symmetric, $\sum_{i=1}^{k} |\lambda_i(\mathscr{E}_2)| \leq \epsilon_2 ||X_{r|k}||_F^2$, and $tr(\mathscr{E}_2) \leq \tilde{\epsilon}_2 ||X_{r|k}||_F^2$.*

  *(iii) The columns of $\mathscr{E}_3$ fall in the column span of $C$ and $tr(\mathscr{E}_3' C^+ \mathscr{E}_3) \leq \epsilon_3^2 ||X_{r|k}||_F^2$.*

  *(iv) The rows of $\mathscr{E}_4$ fall in the row span of $C$ and $tr(\mathscr{E}_4 C^+ \mathscr{E}_4') \leq \epsilon_4^2 ||X_{r|k}||_F^2$.*

*and $\epsilon_1 + \epsilon_2 + \tilde{\epsilon}_2 + \epsilon_3 + \epsilon_4 = \epsilon$, then $\hat{X}$ is a $\epsilon$-approximation embedded matrix for $X$. Specifically, we have $(1 - \epsilon)tr(ZCZ) \leq tr(Z\hat{C}Z) - \min\{0, tr(\mathscr{E}_2)\} \leq (1 + \epsilon)tr(ZCZ)$.*

The proof can be referred to [1]. We then show the following theorem:

**Theorem 2.** *Assume $r > 2k$ and let $V_{2k} \in \mathbb{R}^{d \times r}$ represent $V$ with all but their first $2k$ columns zeroed out. We define $M_1 = V_{2k}'$, $M_2 = \sqrt{k}/||X_{r|k}||_F(X - XV_{2k}V_{2k}')$ and $M \in \mathbb{R}^{(n+r) \times d}$ as containing $M_1$ as its first $r$ rows and $M_2$ as its lower $n$ rows. We construct $R = (Q\Phi)' \in \mathbb{R}^{\tilde{d} \times d}$, which is shown in Algorithm 1 of the main paper. Given $\tilde{d} = \mathcal{O}(\max(\frac{k + log(1/\delta)}{\epsilon^2}, \frac{6}{\epsilon^2 \delta}))$, then for any $X \in \mathbb{R}^{n \times d}$, with a probability of at least $1 - \mathcal{O}(\delta)$, we have*

---

[∗]The first two authors make equal contributions.

(i) $||(RM')'(RM') - MM'||_2 < \epsilon$.

(ii) $|\,||RM_2'||_F^2 - ||M_2'||_F^2| \leq \epsilon k$.

*Proof.* To prove the first result, one can easily check that $M_1 M_2' = \mathbf{0}_{r \times n}$, thus $MM'$ is a block diagonal matrix with an upper left block equal to $M_1 M_1'$ and lower right block equal to $M_2 M_2'$. The spectral norm of $M_1 M_1'$ is 1. $||M_2 M_2'||_2 = ||M_2||_2^2 = \frac{k||X - XV_{2k}V_{2k}'||_2^2}{||X_{r|k}||_F^2} = \frac{k||X_{r|2k}||_2^2}{||X_{r|k}||_F^2}$. As $||X_{r|k}||_F^2 \geq k||X_{r|2k}||_2^2$, we derive $||M_2 M_2'||_2 \leq 1$. Since $MM'$ is a block diagonal matrix, we have $||M||_2^2 = ||MM'||_2 = \max\{||M_1 M_1'||_2, ||M_2 M_2'||_2\} = 1$. $tr(M_1 M_1') = 2k$. $tr(M_2 M_2') = \frac{k||X_{r|2k}||_F^2}{||X_{r|k}||_F^2}$. As $||X_{r|k}||_F^2 \geq ||X_{r|2k}||_F^2$, we derive $tr(M_2 M_2') \leq k$. Then we have $||M||_F^2 = tr(MM') = tr(M_1 M_1') + tr(M_2 M_2') \leq 3k$. Applying Theorem 6 from [2], we can obtain that given $\tilde{d} = \mathcal{O}(\frac{k + log(1/\delta)}{\epsilon^2})$, with a probability of at least $1 - \delta$, $||(RM')'(RM') - MM'||_2 < \epsilon$.

To prove the second result in clear terms, we set $B = M_2'$. Let $B_i$ denote the $i$-th column of $B$, $B_i(b)$ denote the column vector whose $j$-th coordinate is 0 if $h(j) \neq b$, and whose $j$-th coordinate is $B_{j,i}$ if $h(j) = b$.

$$
\begin{aligned}
E_{Q,h}[||RB||_F^2] &= \sum_{i \in [n]} E_{Q,h}[||RB_i||_2^2] \\
&= \sum_{i \in [n]} \sum_{b \in [\tilde{d}]} E_{Q,h}[(\sum_{j:h(j)=b} B_{j,i} Q_{j,j})^2] \\
&= \sum_{i \in [n]} \sum_{b \in [\tilde{d}]} E_h[||B_i(b)||_2^2] = ||B||_F^2
\end{aligned}
\tag{1}
$$

We consider

$$
E_{Q,h}[||RB||_F^4] = \sum_{i \in [n]} E_{Q,h}[||RB_i||_2^4] + \sum_{i \neq z \in [n]} E_{Q,h}[||RB_i||_2^2 ||RB_z||_2^2]
\tag{2}
$$

We bound the first term in Eq.(2) as:

$$
\begin{aligned}
E_{Q,h}[||RB_i||_2^4] &= E_h\Big[\sum_{b \in [\tilde{d}]} E_Q[(RB_i)_b^4] + \sum_{b \neq \tilde{b} \in [\tilde{d}]} E_Q[(RB_i)_b^2] E_Q[(RB_i)_{\tilde{b}}^2]\Big] \\
&= E_h\Big[\sum_{b \in [\tilde{d}]} E_Q[(\sum_{j:h(j)=b} B_{j,i} Q_{j,j})^4] \\
&\quad + \sum_{b \neq \tilde{b} \in [\tilde{d}]} E_Q[(\sum_{j:h(j)=b} B_{j,i} Q_{j,j})^2] E_Q[(\sum_{j:h(j)=\tilde{b}} B_{j,i} Q_{j,j})^2]\Big] \\
&\leq E_h\Big[\sum_{b \in [\tilde{d}]} \Big(\sum_{j:h(j)=b} B_{j,i}^4 + 6 \sum_{j<l:h(j)=h(l)=b} B_{j,i}^2 B_{l,i}^2\Big) + \sum_{b \neq \tilde{b} \in [\tilde{d}]} ||B_i(b)||_2^2 ||B_i(\tilde{b})||_2^2\Big]
\end{aligned}
\tag{3}
$$

For a fixed $b$, we have $E_h[\sum_{j<l:h(j)=h(l)=b} B_{j,i}^2 B_{l,i}^2] \leq \frac{||B_i||_2^4}{\tilde{d}^2}$, so $6 E_h[\sum_{b \in [\tilde{d}]} \sum_{j<l:h(j)=h(l)=b} B_{j,i}^2 B_{l,i}^2] \leq 6/\tilde{d}||B_i||_2^4$. Thus, Eq.(3) can be bounded as:

$$
\begin{aligned}
E_{Q,h}[||RB_i||_2^4] &\leq E_h[||B_i||_4^4] + 6/\tilde{d}||B_i||_2^4 + E_h\Big[\sum_{b \neq \tilde{b} \in [\tilde{d}]} ||B_i(b)||_2^2 ||B_i(\tilde{b})||_2^2\Big] \\
&\leq E_h\Big[\sum_{b \in [\tilde{d}]} ||B_i(b)||_2^4\Big] + 6/\tilde{d}||B_i||_2^4 + E_h\Big[\sum_{b \neq \tilde{b} \in [\tilde{d}]} ||B_i(b)||_2^2 ||B_i(\tilde{b})||_2^2\Big] \\
&\leq (1 + 6/\tilde{d})||B_i||_2^4
\end{aligned}
\tag{4}
$$

Next, for $i \neq z \in [n]$, we bound the second term in Eq.(2) as:

$$
\begin{aligned}
E_{Q,h}[||RB_i||_2^2||RB_z||_2^2] =& E_{Q,h}\Big[\sum_{b\in[\tilde{d}]}\Big(\sum_{j:h(j)=b}B_{j,i}Q_{j,j}\Big)^2\Big(\sum_{l:h(l)=b}B_{l,z}Q_{l,l}\Big)^2\Big] \\
&+E_{Q,h}\Big[\sum_{b\neq\tilde{b}\in[\tilde{d}]}\Big(\sum_{j:h(j)=b}B_{j,i}Q_{j,j}\Big)^2\Big(\sum_{l:h(l)=\tilde{b}}B_{l,z}Q_{l,l}\Big)^2\Big] \\
=& E_{Q,h}\Big[\sum_{b\in[\tilde{d}]}\Big(\sum_{j:h(j)=b}B_{j,i}^2Q_{j,j}^2+\sum_{j\neq l:h(j)=h(l)=b}B_{j,i}B_{l,i}Q_{j,j}Q_{l,l}\Big) \\
&\Big(\sum_{j:h(j)=b}B_{j,z}^2Q_{j,j}^2+\sum_{j\neq l:h(j)=h(l)=b}B_{j,z}B_{l,z}Q_{j,j}Q_{l,l}\Big)\Big] \\
&+E_{Q,h}\Big[\sum_{b\neq\tilde{b}\in[\tilde{d}]}\Big(\sum_{j:h(j)=b}B_{j,i}Q_{j,j}\Big)^2\Big(\sum_{l:h(l)=\tilde{b}}B_{l,z}Q_{l,l}\Big)^2\Big] \\
=& E_{Q,h}\Big[\sum_{b\in[\tilde{d}]}\Big(\sum_{j\neq l:h(j)=h(l)=b}B_{j,i}B_{l,i}Q_{j,j}Q_{l,l}\Big) \\
&\Big(\sum_{j\neq l:h(j)=h(l)=b}B_{j,z}B_{l,z}Q_{j,j}Q_{l,l}\Big)\Big] \\
&+E_h\Big[\sum_{b\in[\tilde{d}]}||B_i(b)||_2^2||B_z(b)||_2^2+\sum_{b\neq\tilde{b}\in[\tilde{d}]}||B_i(b)||_2^2||B_z(\tilde{b})||_2^2\Big] \\
=& 4E_h\Big[\sum_{b\in[\tilde{d}]}\sum_{j<l:h(j)=h(l)=b}B_{j,i}B_{l,i}B_{j,z}B_{l,z}\Big]+||B_i||_2^2||B_z||_2^2 \\
\leq& 4E_h\Big[\sum_{b\in[\tilde{d}]}\langle B_i(b),B_z(b)\rangle^2\Big]+||B_i||_2^2||B_z||_2^2 \\
\leq& 4E_h\Big[\sum_{b\in[\tilde{d}]}||B_i(b)||_2^2||B_z(b)||_2^2\Big]+||B_i||_2^2||B_z||_2^2 \\
\leq& 4/\tilde{d}\sum_{j,l\in[d]}B_{j,i}^2B_{l,z}^2\Big]+||B_i||_2^2||B_z||_2^2 \\
=& (1+4/\tilde{d})||B_i||_2^2||B_z||_2^2
\end{aligned}
\tag{5}
$$

Combining Eqs.(1), (2), (4) and (5), we bound the variance of $||RB||_F^2$, which is denoted by $Var(||RB||_F^2)$, as:

$$
\begin{aligned}
Var(||RB||_F^2) =& E_{Q,h}[||RB||_F^4]-(E_{Q,h}[||RB||_F^2])^2 \\
\leq& \sum_{i\in[n]}(1+6/\tilde{d})||B_i||_2^4+\sum_{i\neq z\in[n]}(1+4/\tilde{d})||B_i||_2^2||B_z||_2^2-||B||_F^4 \\
\leq& 6/\tilde{d}||B||_F^4
\end{aligned}
\tag{6}
$$

Using Chebyshev's inequality, we arrive at

$$
P(|||RB||_F^2-||B||_F^2|\geq\epsilon||B||_F^2)\leq\frac{Var(||RB||_F^2)}{\epsilon^2||B||_F^4}\leq\frac{6}{\epsilon^2\tilde{d}}
\tag{7}
$$

By setting $\delta=\frac{6}{\epsilon^2\tilde{d}}$, we can obtain that with a probability of at least $1-\delta$, $|||RB||_F^2-||B||_F^2|\leq\epsilon||B||_F^2$. As $||B||_F^2\leq k$, we complete the proof of the second result. $\square$

Table 1: Data sets used in the main paper.

| Data Set | # INSTANCE | # FEATURES | # CLASSES |
|---|---|---|---|
| COIL20 | 1440 | 1024 | 20 |
| SECTOR | 9619 | 55,197 | 105 |
| RCV1 | 534,135 | 47,236 | 53 |
| ILSVRC2012 | 1,331,167 | 4,096 | 1,000 |

Figure 1: Illustration of cluster samples from ILSVRC2012 data set that are generated by the proposed SE. Each row illustrates several representative images of one cluster.

## 2  Data Sets

This paper evaluates the performance of the proposed method on four real-world data sets: COIL20, SECTOR, RCV1 and ILSVRC2012. The COIL20 [3] and ILSVRC2012 [4] data sets are collected from website[2][3], and other data sets are collected from the LIBSVM website[4].

## 3    Case Study

This section presents a case study in which the proposed SE is applied to a large-scale image clustering application. Figure 1 shows some sample clusters from ILSVRC2012 that are generated by SE. Each row illustrates several representative images of one cluster. We observe from this figure that similar images are well clustered. This case study suggests that SE works well in practical large-scale clustering applications.

## Footnotes

[2]http://www.cs.columbia.edu/CAVE/software/softlib/coil-20.php

[3]http://www.image-net.org/challenges/LSVRC/2012/

[4]https://www.csie.ntu.edu.tw/ cjlin/libsvmtools/datasets/