[Reviews · NeurIPS 2017]

Reviewer 1



This paper proposes a dimension reduction method for k-means clustering, and provides a theoretical guarantee of the proposal. This paper is overall well written. In particular, the theoretical contribution is interesting and solid. However, although the proposed method is empirically shown to be superior to other advanced clustering methods, the results are not reliable due to the following reasons: - To what values \epsilon and \delta are set in experiments? Since they work as parameters in practice, such information is crucial in empirical evaluation. - Related to the above point, how to set them in real applications? Is there any guideline? - In addition, the sensitivity of the clustering performance of the proposed method with respect to changes in such parameters should be empirically examined. - What is the measure of clustering results? Accuracy is not appropriate and variation of information should be used instead.

Reviewer 2



The authors proposed a simply but effective k-means based clustering algorithm with a new random projection dimensionality reduction technique. Both theoretical and empirical results show the effectiveness and efficiency of the proposed method compared to the baseline k-means and comparison dimensionality reduction techniques. The paper would be a good fit to NIPS and proposed method would be useful for the public community. Here are some minor points. 1. Line 100 and 101 are confusing, local optimal or global optimal? 2. Practically, how to set epsilon and delta? Does it effect the results? 3. Line 217 and 218 the authors said that SE outperforms RP regarding accuracy, which can not be seen from Figure 1. 4. The runtime for the proposed method is O(nnz(X)), but for the evaluated image datasets nnz(X) is closer to n *d, right? However, the proposed method is still much faster than RP with complexity of O(nd). Maybe the authors can explain more about this. 5. Is that possible to extend the method to the kernel space?

Reviewer 3



The authors proposed a sparse embedded k-means clustering algorithm to improve the running time of matrix multiplication of current proposed randomization projection method under sparse setting of data matrix in the literature. In particular, they demonstrated that their algorithms achieve the calculation time of matrix multiplication of order proportional to the number of non-zeroes entries of data matrix. I think the sparse embedded k-means clustering algorithm is rather interesting; however, it is not a very surprising improvement given the current results from the paper of C. Boutsidis et al. (2015). More specifically, both papers try to approximate low dimension solution for the original solution of K-means problem. To do that, C. Boutsidis et al. (2015) proposed to multiply the data matrix X with a random matrix having entries $1/\sqrt{d'}$ or $-1/\sqrt{d'}$ where $d'$ is the dimension of the approximation solution. However, their proposed solution does not work work well with sparse setting of data matrix as the calculation of matrix multiplication from their method greatly depends on the number of data points and the number of clusters. To account for that drawback, the authors of current paper proposed to multiply the data matrix with sparse random matrix inspired from diagonal Rademacher matrix. At the high level, it is unsurprisingly will improve the computation of matrix multiplication from C. Boutsidis et al.'s work under sparse setting of data matrix. At the setting of rather dense data matrix, I think both methods may differ slightly in terms of accuracy and matrix multiplication time. In summary, even though the paper is rather interesting, I think the contribution of the paper is not very surprising. Some minor comments: (1) Will the condition in Definition 2 hold for any matrix $D$? (2) It may be better for readers if the authors replace notation $\widehat{D}$ in Lemma 1 with less confusing notation. At the moments, the notations in Lemma 1 look rather similar. (3) I do not see the clustering accuracy with Random projection method from C. Boutsidis et al.'s work with RCV1 data in the experiment section. I wonder what may happen with that accuracy result?